# Semi-Quantitative Evaluation of Asymmetricity of Dialysis Membrane Using Forward and Backward Ultrafiltration

**DOI:** 10.3390/membranes12060624

**Published:** 2022-06-15

**Authors:** Akihiro C. Yamashita, Toshiki Kakee, Takahisa Ono, Jun Motegi, Satoru Yamaguchi, Takashi Sunohara

**Affiliations:** 1Department of Chemical Science and Technology, Faculty of Bioscience and Applied Chemistry, Hosei University, Tokyo 184-8584, Japan; toshiki.kakee0714@gmail.com (T.K.); t-ono@nomura-nms.co.jp (T.O.); 2Nipro Co., 3-9-3 Honsho-Nishi, Kita-ku, Osaka 531-8510, Japan; moteki-jun@nipro.co.jp (J.M.); yamaguchi-satoru@nipro.co.jp (S.Y.); sunohara-takashi@nipro.co.jp (T.S.)

**Keywords:** dialysis, membrane, asymmetricity, homogeneous, ultrafiltration

## Abstract

Performance of the dialysis membrane is strongly dependent upon the physicochemical structure of the membrane. The objective of this study is to devise a new in vitro evaluation technique to quantify the physicochemical structures of the membrane. Three commercial dialyzers with cellulose triacetate (CTA), asymmetric CTA (termed ATA^®^), and polyether sulfone (PES) membranes (Nipro Co., Osaka, Japan) were employed for investigation. Forward and backward ultrafiltration experiments were performed separately with aqueous vitamin B_12_ (MW 1355), α-chymotrypsin (MW 25,000), albumin (MW 66,000) and dextran solutions, introducing the test solution inside or outside the hollow fiber (HF), respectively. Sieving coefficients (s.c.) for these solutes were measured under the test solution flow rate of 200 mL/min and the ultrafiltration rate of 10 mL/min at 310 K, according to the guidelines provided by Japanese academic societies. We defined the ratio of s.c. in the backward ultrafiltration to that in the forward ultrafiltration and termed it the index for asymmetricity (IA). The IA values were unity for vitamin B_12_ and α-chymotrypsin in all three of the dialyzers. The IA values for albumin, however, were 1.0 in CTA, 1.9 in ATA^®^, and 3.9 in PES membranes, respectively, which corresponded well with the fact that CTA is homogeneous, whereas ATA^®^ and PES are asymmetrical in structure. Moreover, the asymmetricity of ATA^®^ and PES may be different by twofold. This fact was verified in continuous basis by employing dextran solution before and after being fouled with albumin. These findings may contribute to the development of a novel membrane for improved success of dialysis therapy.

## 1. Introduction

Many commercial dialyzers are available these days, among which are the mainstream ones that include synthetic polymeric membranes made from petroleum [1] for the following two reasons: (1) dialyzers with these polymeric membranes have higher clearances for large solutes such as β_2_-microglobulin that was reported to be one of major factors in developing carpal tunnel syndrome in dialysis patients [2], or it is known that classic regenerated cellulosic membrane cannot remove these large substances effectively [3]; and (2) dialyzers with these polymeric membranes generally show better biocompatibility than the classic cellulosic ones [4,5]. Although there are many choices [6], there is no rule for choosing one from the many products, except in the case of patients with severe bio-incompatible reactions found with a certain kind of membrane, such as classic cuprammonium cellulose due to the hydroxyl groups in the chemical structure [7], so-called “first use reaction” [8], or generation of bradykinin due to the use of angiotensin-converting enzyme inhibitors (ACE-I) with a highly negatively charged membrane, AN69^®^ [9], etc. Solute removal performances as well as water permeabilities, however, must be taken into consideration for reducing clinical symptoms [10,11]. For this purpose, the treatment modality was shifting from conventional hemodialysis (HD) to on-line hemodiafiltration (HDF) with post-dilution substitution in European countries [12,13,14], and with pre-dilution substitution in Japan [15,16]. To perform safe and high-efficiency on-line HDF with either post-dilution or pre-dilution, performance of the membrane, closely related to its physicochemical structures, plays a significant role. No relevant technique, however, has been reported to characterize the membrane structure quantitatively beyond the direct observation by using scanning electron microscope (SEM) and the use of mathematical models [17,18,19]. Moreover, since the performance of the dialyzer, especially for clearances of large solutes, is normally decreasing drastically during the treatment, we must pay a great deal of attention to the membrane fouling due to the adsorption of plasma proteins [20,21,22]. Then, the objective of this study is to devise a new in vitro evaluation technique to evaluate physicochemical structures of the membrane semi-quantitatively, before and after being fouled with aqueous albumin solution.

## 2. Materials and Method

Three commercial dialyzers with cellulose triacetate (CTA), asymmetric CTA (termed ATA^®^), and polyether sulfone (PES) membranes (all three devices are produced and distributed by Nipro Co., Osaka, Japan) were employed for investigation. These devices have the same surface area of 2.1 m^2^ and are tabulated in Table 1 with their commercial names and abbreviations. Among those commercial devices, FB (CTA) is a so-called “super high-flux” or medium cut-off (MCO) [23] dialyzer, whereas FIX (ATA^®^) and MFX (PES) are also super high-flux but are diafilters that are specifically designed for HDF treatment. Forward (In-to-Out) ultrafiltration experiments were performed by introducing the test solution inside the hollow fiber (HF) and the ultrafiltrate taken from outside the HF with aqueous vitamin B_12_ (MW 1355, FUJIFILM Wako Pure Chemical Co., Osaka, Japan), α-chymotrypsin (MW 25,000, bovine pancreas, Sigma-Aldrich, St. Louis, MO, USA), and albumin (MW 66,000, bovine serum, FUJIFILM Wako Pure Chemical Co.) solutions (Figure 1). Backward (Out-to-In) ultrafiltration experiments were devised by introducing the test solution outside the HF and the ultrafiltrate from inside the HF, respectively (Figure 2). Experiments with aqueous dextran (Sigma-Aldrich, St. Louis, MO, U.S.A.) solution in the wide range of MW were also performed separately. Sieving coefficients (s.c.) for those solutes were measured under the test solution flow rate *Q*_B_ = 200 mL/min and the ultrafiltration flow rate *Q*_F_ = 10 mL/min at 310 K, in accordance with the Guidelines from the Japanese Society for Artificial Organs [24] and from the Japanese Society for Dialysis Therapy [25]. All of the investigated devices (dialyzers/diafilters) were used only once and all of the experiments were repeated at least three times, by using separate devices of the same lot.

In order to foul the membrane, aqueous albumin solution (2.0 mg/mL) was prepared with a phosphate buffer at pH = 7.4 and was pumped into a dialyzer/diafilter at 200 mL/min and ultrafiltration was executed at 100 mL/min for 60 min, returning the ultrafiltrate back to the original albumin solution tank. After rinsing, the albumin was immobilized in or on the membrane by using 2.0% glutaraldehyde (GA, FUJIFILM Wako Pure Chemical Co., Osaka, Japan) solution, in the way that we have previously reported [22], mimicking the clinical fouling due to plasma proteins (Figure 3). Then, the same ultrafiltration experiments with the aqueous dextran solution were performed to compare dynamic changes of solute transport after being fouled. Most of the chemicals used in the experiments are tabulated in Table 2. All the experiments were repeated at least three times and results were shown in a mean ± S.D.

All the transport experiments were performed aqueous in vitro with ion-exchanged water produced from tap water with two prefilters in series. The test solution included a single component (vitamin B_12_, α-chymotrypsin, or albumin) except for the dextran that was a mixture of five commercial chemicals with a wide variety of molecular weights (MW), covering MW from around 1000 to 200,000 (Table 2). A phosphate buffer solution at pH 7.4 was used as a solvent when a protein (α-chymotrypsin or albumin) was chosen as a test solute and the ion-exchanged water was used as the solvent for the vitamin B_12_ and dextran. Initial concentrations of the test solutes were also tabulated in Table 2.

Concentrations for vitamin B_12_, α-chymotrypsin, and albumin were directly measured by using an ultraviolet-visible spectrophotometer (UV-1280, Shimadzu Co., Kyoto, Japan) with the wavelength of 260, 282, and 278 nm, respectively, for these solutes. Absorbance of the sample was measured and was converted into the concentration by using a pre-determined calibration curve, a linear relationship between the absorbance and the concentration. Concentration for the dextran was measured by using a gel permeation chromatography ((GPC); System: 1120 Compact LC, Tosoh Co., Tokyo, Japan; column: GF-510HQ, Showa Denko Co., Tokyo, Japan) with the flow rate of 0.5 mL/min. The mobile phase was distilled water and the volume of injected sample with no pretreatment from the automatic sampler was 100 µL. The refractive index for the differential refractometer was 64.0 × 10^−3^ RIU/FS. The calibration curve was made using a classic method in which the retention time of eight dextran chemicals with monodispersed MW was plotted against the MW and it was fitted with a cubic spline.

## 3. Theoretical

We used the sieving coefficient, s.c., as an index for permeation of the solute of interest in the ultrafiltration experiments. There are, however, many definitive equations of s.c., including: (1) the ratio of concentration in the filtrate to that in the blood, s.c._1_ [26]; (2) the ratio of concentration in the filtrate to the arithmetic mean of the concentration at the blood inlet and outlet, s.c._2_ [26]; and (3) the equation derived by integrating the material balance in the infinitesimally small portion of the ultrafilter, considering non-linear concentration distribution in the flow direction of the device, s.c._3_ [27]. Although equations for s.c._1_ and s.c._2_ are simple, the values from s.c._1_ are always greater than those from s.c._3_, while those of s.c._2_ are always a little smaller than s.c._3_. In addition to that, the equation for s.c._3_ is a little cumbersome to use and it returns unstable values when the value is close to unity. Then, we proposed a new definitive equation of the sieving coefficient in a simple way, using a geometric mean of the concentration at the blood inlet and outlet in the denominator of the definition and termed it as s.c._4_ [28,29], i.e.:(1)$$s.c._{4}=\frac{C_{F}}{\sqrt{C_{\mathrm{Bi}}\times C_{\mathrm{Bo}}}}$$
where *C*_Bi_ and *C*_Bo_ are the concentrations in blood at the inlet and outlet of the dialyzer/diafilter (mg/mL), and *C*_F_ is the concentration in the ultrafiltrate (mg/mL). This definition is simple yet returns very close values to those from s.c._3_ and never returns unstable values when applied to clinical and/or experimental data, even when the value is close to unity [29].

The ratio of s.c._4_ in the backward ultrafiltration to that in the forward ultrafiltration was defined for evaluation of asymmetricity (heterogeneity) of the dialysis membrane and we termed it the index for asymmetricity (IA), i.e.:(2)$$\mathrm{IA}=\frac{s.c._{4}\;\mathrm{under}\;\mathrm{backward}\;\mathrm{filtration}}{s.c._{4}\;\mathrm{under}\;\mathrm{forward}\;\mathrm{filtration}}$$

The value of IA is expected to be unity for most solutes in the homogeneous membrane, while it should be greater than unity for molecules greater than a certain size in the asymmetric membranes, according to the degree of asymmetricity (heterogeneity).

After performing ultrafiltration of albumin solution for 60 min, albumin was immobilized on or in the membrane, mimicking the clinical situation of fouling. Under such circumstances, the pore diameters may become smaller than those before fouling. Then, the following index was defined and termed the index for fouling, IF, to evaluate the degree of fouling, i.e.:(3)$$\mathrm{IF}=\left(1-\frac{s.c._{4}\;\mathrm{after}\;\mathrm{fouling}}{s.c._{4}\;\mathrm{before}\;\mathrm{fouling}}\right)$$

## 4. Results and Discussion

According to our previous FE-SEM observation of these membranes [30], not much difference was found between inside and outside the membrane in the CTA that implied that it was a homogeneous membrane. We found, however, large macro pores outside the ATA^®^ and PES membranes with no macro pores inside that proved that these membranes had asymmetric or heterogeneous structures. As shown in Figure 4, values of s.c._4_ in the homogeneous membrane are expected to be identical under forward and backward filtrations, because the diameters of the pores are thought to be uniform across the membrane; however, those in the asymmetric membranes should not be identical because the diameters of the pores are greater outside, and much smaller inside, especially those in the skin layer that is the region of 1–2 µm from the inside surface.

The time courses of s.c._4_ for albumin were shown in Figure 5, showing forward ultrafiltration (left) and backward ultrafiltration (right). All of the values are represented as mean ± standard deviation (S.D.), and data are well reproduced with fairly small S.D.s. Comparing a pair of these schemas, we found that values of s.c._4_ in backward ultrafiltration were much higher than those in forward ultrafiltration in the asymmetric membranes (ATA^®^ (FIX) and PES (MFX)), while they were almost identical in the homogeneous membrane (CTA (FB)) that corresponded well with the direct observations by FE-SEM [30] discussed above. Results with CTA and PES in the forward ultrafiltration show almost the same s.c._4_ value of 0.05; however, they were different by four-fold in the backward experiments, which clearly showed the different characteristics of these two membranes.

According to the classic mass transfer theory [17,18,19], the s.c._4_ in the forward ultrafiltration and in the backward ultrafiltration are expected to be identical, and the values of IA were indeed unity for vitamin B_12_ and α-chymotrypsin in all three investigated devices (Figure 6). However, unlike these solutes, IA values for albumin were varied from 1.0 in CTA, 1.9 in ATA^®^, to 3.9 in PES membranes, respectively. These findings include the following two things: (1) CTA is the only homogeneous membrane of the three; (2) the asymmetricity (heterogeneity) of ATA^®^ and PES may be different by twofold (= 3.9/1.9). We already reported the similar results in the polysulfone membrane dialyzers with vitamin E coating by performing forward and backward ultrafiltration experiments [31]. We also reported on the dialysis experiments, in which clearances for small solutes in forward and backward directions were identical regardless of membrane structures and higher clearances were found in backward dialysis for large solutes in the dialyzers with the asymmetric membrane, employing eleven commercial dialyzers with different kinds of membranes [30]. The results demonstrated here were consistent with the findings previously reported.

Figure 7 shows the cut-off curves taken by using dextran solution in intact ATA^®^ membrane diafilter (FIX) for both forward (left) and backward (right) ultrafiltration. Values of s.c._4_ were essentially zero at the MW of 52,000 in forward and 56,000 in backward ultrafiltration experiments, respectively. However, few changes were found with time during the experiments for 240 min, except for the data taken 5 min after starting the experiments that may be caused by the dilution of previously loaded ion-exchanged water. Then, we concluded that no fouling may be expected with low concentration dextran solution even under ultrafiltration of 10 mL/min. Since similar results were found with the other two devices (FB (CTA), MFX (PES)), arithmetic mean values from 15 to 240 min are represented in all three devices hereafter.

Figure 8 shows the comparison of s.c._4_ between the intact and albumin-immobilized (fouled) devices under forward ultrafiltration. Curves relating to the fouled membranes were shifted to the left in all three membranes; moreover, much greater differences were found between the intact and albumin-immobilized PES (MFX) membrane than the other two membranes. The same data were evaluated by using IA as an index in Figure 9 that verified the results found in Figure 6 on a continuous basis. The value of IA in the PES (MFX) dialyzer started increasing from MW around 10,000, and it eventually exceeded 3.0 and 6.0 at MW around 40,000 in the intact and in the albumin-immobilized membrane, respectively. This fact implies that the membrane fouling reduces the pore diameter. The highest values of IA in ATA^®^ (FIX) and CTA (FB) membranes were much smaller than corresponding values found in PES (MFX), which concluded that the degree of membrane fouling was less in these two dialyzers. The asymmetricity of the membrane is the highest in PES and the lowest in CTA.

After being fouled, the pore size of the membrane should get smaller than the original size, which decreases the forward transport of large solutes but not the backward transport, because once the molecule of interest penetrates the pore either from inside or from outside, it should go through. Therefore, the mechanism of the increased asymmetricity in membranes may be caused by the reduced forward transport, leaving the backward transport nearly unchanged.

Figure 10 shows the relationship between IF and MW. As shown in Figure 8, although PES (MFX) had a higher solute permeability than CTA (FB), it also showed the highest value of IF or the highest reduction in transport, especially for the solutes whose MW ranged from 10,000 to 40,000, which included important large marker substances to be removed, such as β_2_-microglobilin (MW 11,800), prolactin (MW 23,000), FGF-23 (MW 32,000), and α_1_-microglobulin (MW 33,000) [32]. Moreover, ATA^®^ (FIX) had the highest solute permeability (Figure 8); also, it showed the lowest reduction rate in permeability in the MW ranging less than 40,000. This fact correlated well with the rates reported from other basic [33] and clinical studies [34,35], suggesting less reduction in solute permeability in ATA^®^ (FIX) during the treatment than other commercial devices.

There are some limitations in this study. Since all of the experiments were performed with dilute aqueous system in vitro, absolute values of sieving coefficients for the solute of interest may or may not be realistic compared with the values obtained clinically. Moreover, the value of IA was not identical between when the test proteins were used (Figure 6) and when the dextran was used (Figure 9) for the same MW. This is probably caused by the differences of molecular structure of the solute of interest, rather than MW. In addition, little is considered for adsorption by the membrane [36,37] because investigated commercial devices are known to have limited adsorption characteristics in relation to the test solutes. Future study may be necessary, employing membranes with adsorption characteristics to clarify whether a similar classification is possible by using the concept of IA.

## 5. Conclusions

Performance of the dialysis membrane, the most important part of the dialyzer, is strongly dependent upon the physicochemical structure of the membrane. The rate of solute transport across the membrane in forward and backward directions are identical for small solutes, regardless of the membrane structure. A wedge-like pore size structure in the radial direction in the asymmetric dialysis membrane, however, was semi-quantitatively evaluated by performing forward and backward ultrafiltration experiments, introducing a new index, IA. A degree of fouling after being fouled by albumin was also semi-quantitatively evaluated by performing the same forward and backward ultrafiltration experiments, introducing a new index, IF. These findings may contribute to the development of a novel membrane for the further success of the dialysis therapy.

## Figures and Tables

**Figure 1 membranes-12-00624-f001:**
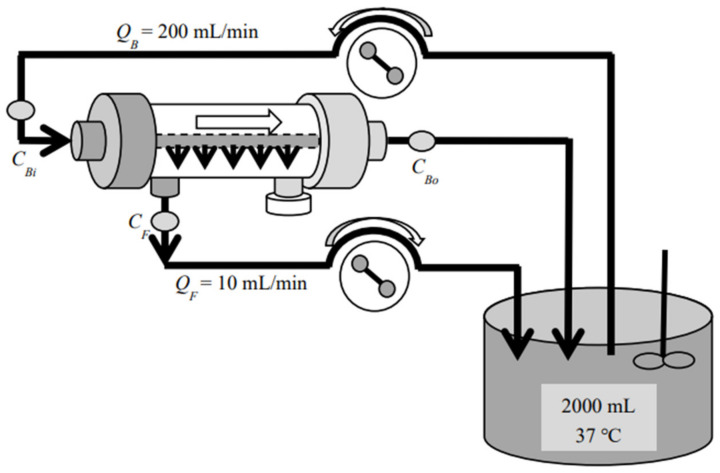
Test circuit for the forward (In-to-Out) ultrafiltration experiment.

**Figure 2 membranes-12-00624-f002:**
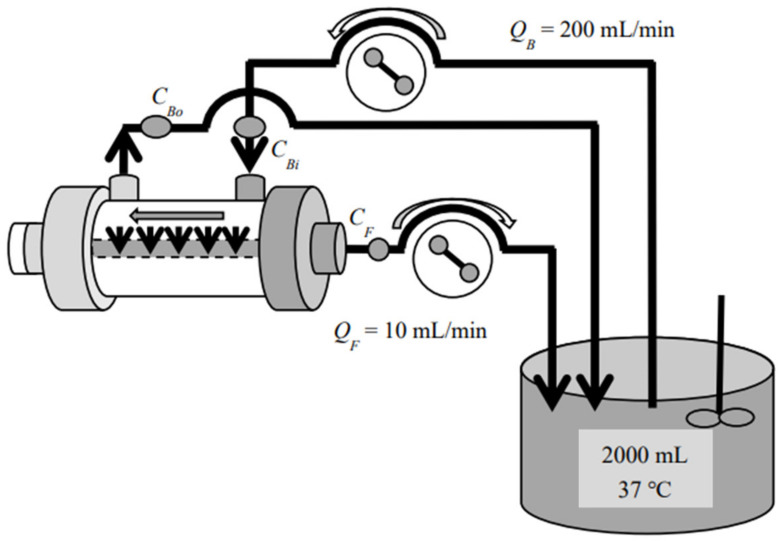
Test circuit for the backward (Out-to-In) ultrafiltration experiment.

**Figure 3 membranes-12-00624-f003:**
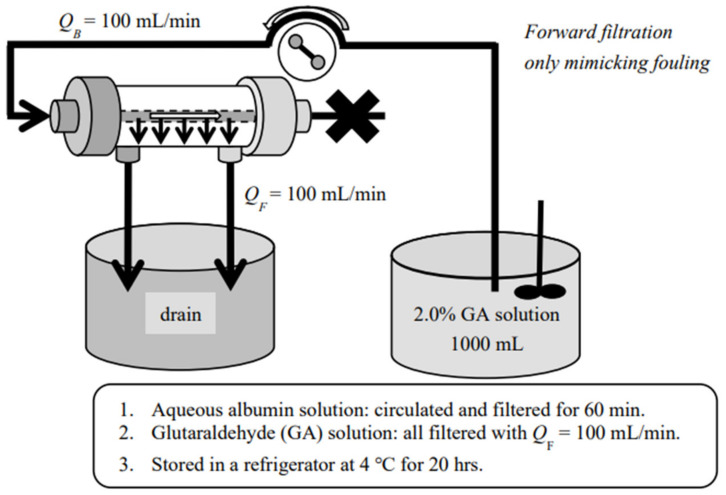
Immobilization of albumin on the dialysis membrane.

**Figure 4 membranes-12-00624-f004:**
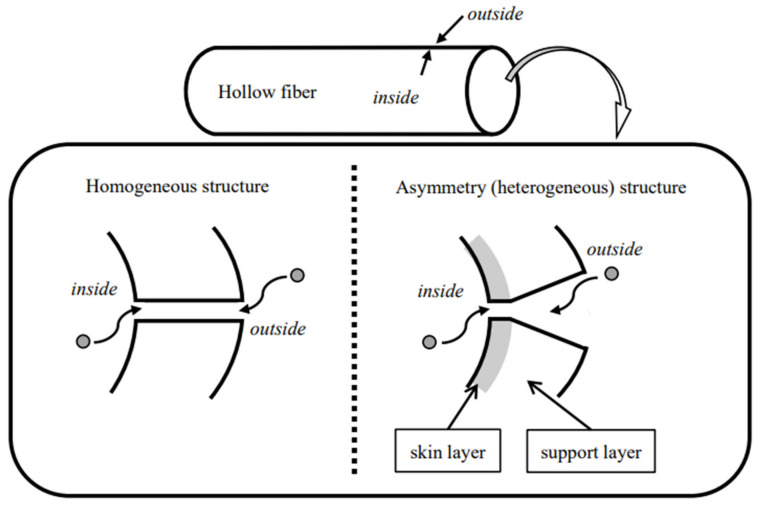
Physicochemical structures of the dialysis membrane.

**Figure 5 membranes-12-00624-f005:**
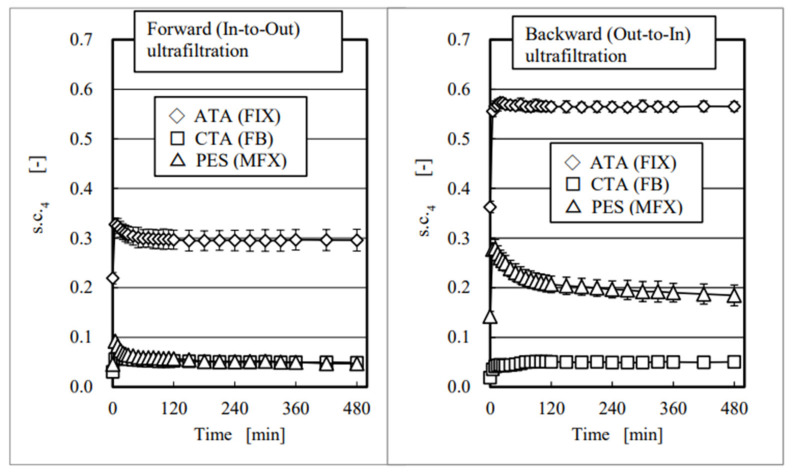
Time courses of the sieving coefficient for albumin in three investigated modules.

**Figure 6 membranes-12-00624-f006:**
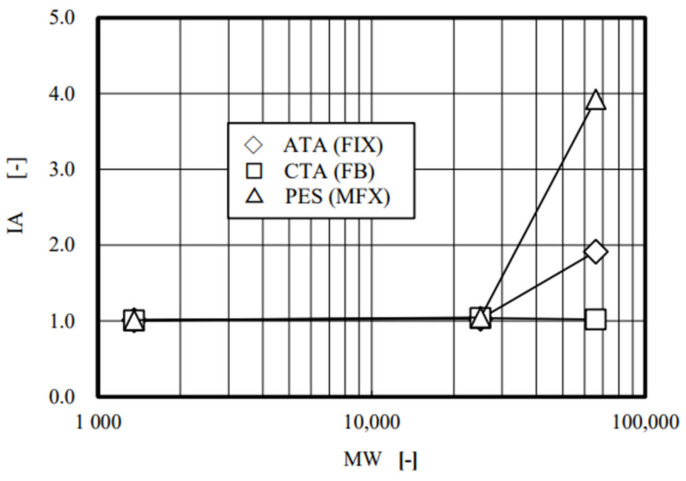
Relationship between the index for asymmetricity (IA) and molecular weight of the solute (MW).

**Figure 7 membranes-12-00624-f007:**
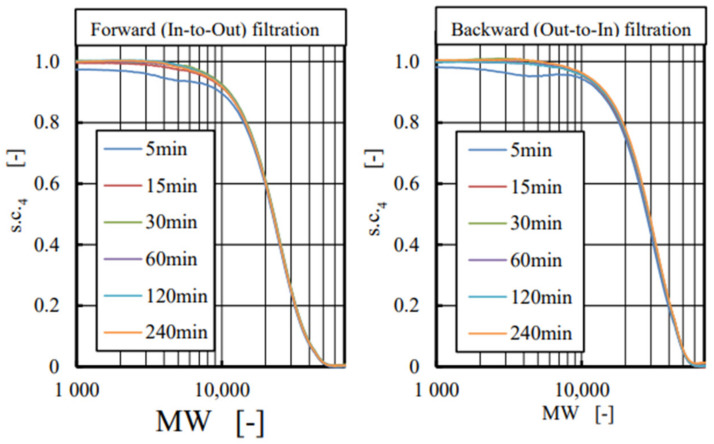
Relationship between the sieving coefficient (s.c._4_) and MW in intact ATA^®^ membrane module (FIX) using dextran with wide variety of MW. No changes were found with time.

**Figure 8 membranes-12-00624-f008:**
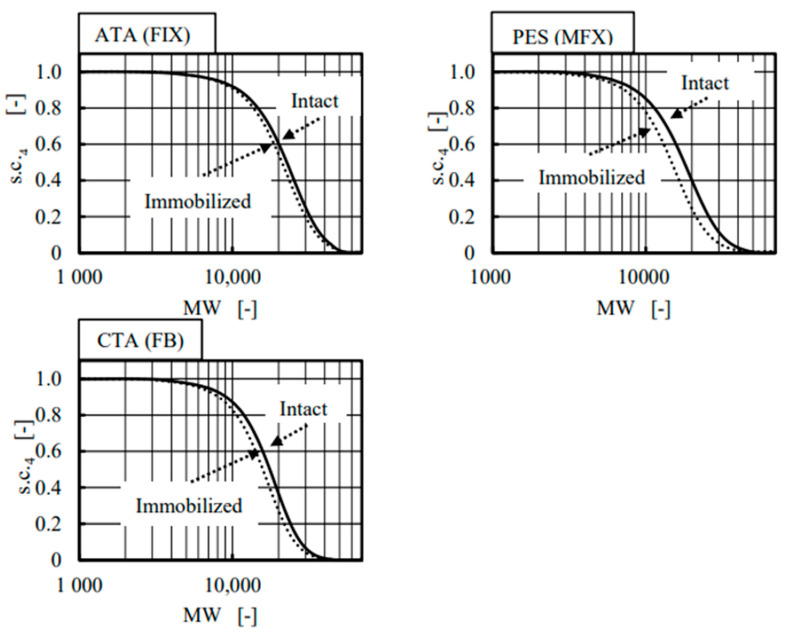
Comparison of cut-off curves between intact and immobilized membrae under forward (In-to-Out) filtration using dextran with wide variety of MW.

**Figure 9 membranes-12-00624-f009:**
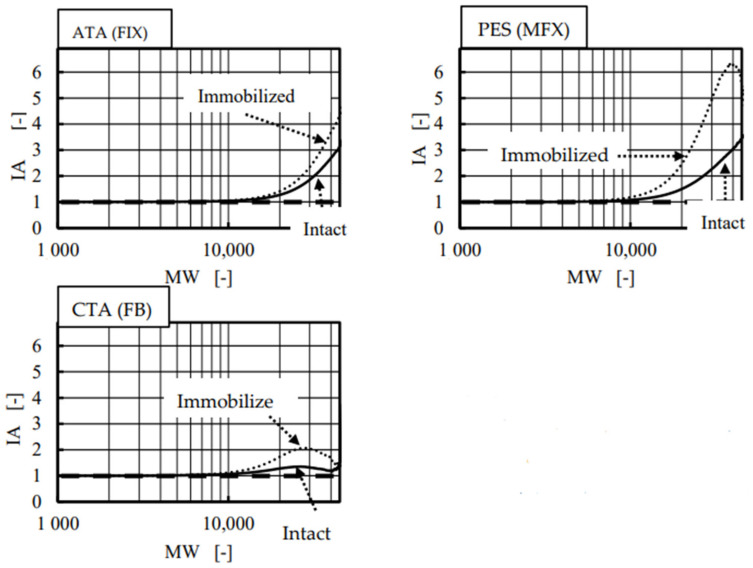
Relationship between the index for asymmetricity (IA) and MW using dextran with wide variety of MW before and after immobilization of albumin.

**Figure 10 membranes-12-00624-f010:**
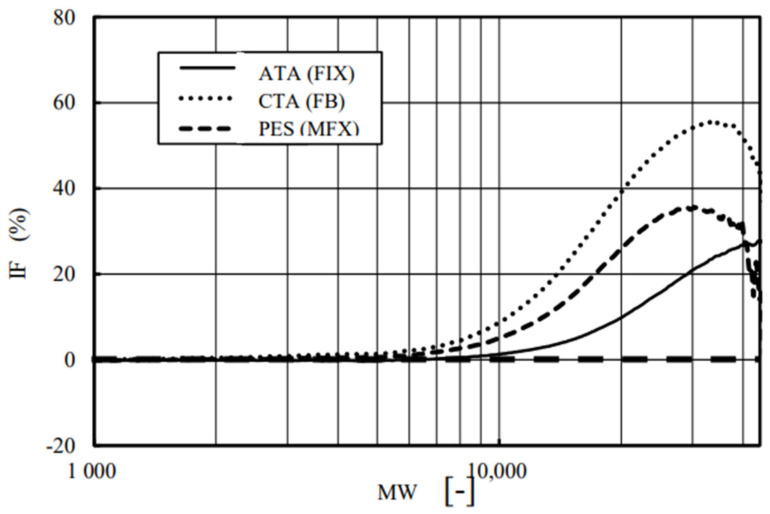
Relationship between the index for fouling (IF) and MW. Forward (In-to-Out) filtration (*n* = 3).

**Table 1 membranes-12-00624-t001:** Three investigated dialyzers.

Commercial Name	Membrane	Physicochemical Structure of the Membrane	Abbreviations of the Commercial Name
FIX-210Seco	CTA (ATA^®^)	Asymmetry	FIX
FB-210UHβeco	CTA	Homogeneous	FB
MFX-21Seco	PES	Asymmetry	MFX

CTA: Cellulose triacetate; PES: Polyether sulfone; ATA: Asymmetric cellulose triacetate. All three devices have the same membrane surface area of 2.1 m^2^ are sterilized by γ-ray are produced and distributed by Nipro Co., Osaka, Japan.

**Table 2 membranes-12-00624-t002:** Chemicals used in the experiments.

Solutes	Molecular Weight (-)	Produced	Purpose	Initial Concentration (mg/mL)
vitamin B_12_	1355	FUJIFILM Wako Pure Chemical Co., Osaka, Japan.	Test solute	0.025
α-chymotripsin ^1^	25,000	Sigma-Aldrich, St. Louis, MO, USA.		0.305
albumin ^1^	66,000	FUJIFILM Wako Pure Chemical Co., Osaka, Japan.	Test solute & Foulant ^2^	24.0
dextran ^3^	~1500	Sigma-Aldrich, St. Louis, MO, USA	Test solute	0.50 ^4^
	~25,000			
	~40,000			
	~60,000			
	~200,000			

^1^ Test solutions were prepared with phosphate buffer solution (PBS) of pH = 7.4; ^2^ A 2.0% glutaraldehyde (GA) solution was used for immobilization of albumin; ^3^ Concentrations of dextran were measured by a gel permeation chromatography (GPC); ^4^ A 1.0 g of all five dextran chemicals with different MW were dissolved in 2000 mL of the ion-exchanged water.

## Data Availability

Not applicable.

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
