# Peer review of "Semi-Quantitative Evaluation of Asymmetricity of Dialysis Membrane Using Forward and Backward Ultrafiltration"

_membranes, 2022, doi:10.3390/membranes12060624_

Round 1
Reviewer 1 Report
The present manuscript describes research developed as a continuation of previous work done by ACY team, namely cited references [25] and [26]. In the earlier study, polysulfone membranes were investigated in their asymmetric behaviour regarding ultrafiltration sieving coefficients of key molecules; in the latter, symmetrical and homogeneous membranes of different materials were compared and a dialysis asymmetry factor, based on solute clearance ratio (reverse:normal) was defined and compared among many dialysis modules.
The development of better parameters to evaluate the asymmetry of sieving behaviour of commercial dialysers seems to be the intent of the study in the present manuscript. Three hollow fibre devices were selected on the bases of membrane material and membrane structure and evaluated.
The study is interesting but the manuscript needs more input before reaching publication stage. Please find below the main comments to this manuscript.
General comments:
Why did the authors decide to change the asymmetry evaluation based on solute clearance ratio, and presently look to backward:forward filtration sieving coefficient ratio? Does the latter parameter provide better insight on asymmetric transport behaviour?
There is a trend in the development of dual-skin hollow fibre membranes for haemodialysis (seen in, e.g., DOI: 10.1046/j.1523-1755.2000.00231.x, and DOI: 10.1016/j.aej.2022.03.043) with the purpose of gaining significant improvements in the forward transport or retention of key molecules while impairing the reverse transport of endotoxins. In light of this this trend, would it still be worth to study the membrane asymmetry parameters?
Abstract, line 23 and 30 – Specific operating conditions in the study are not what the potential readers need to know in the abstract. However, if the authors find it important, should write it in words, as the symbols used to the operating variables were not yet explained. The term “continuous” is misleading in this context and should be avoided.
Materials and Methods – This section must be improved. Which was the quality of the water used in the preparation of solutions? Reference to the use of buffer only comes in the discussion section. Buffers were used in all or just some solutions? Which ones and for which pH values?
Were the solutions mainly binary mixtures (solute and water) or there were multicomponent mixtures of the target solutes? Which precise set of molecular dextrans were used in this work? Which was the concentration of each solute in the prepared solutions?
Were the test-solutions microfiltered or centrifuged at high G-force, previously to the test-runs, to eliminate possible microparticles (impurities, microbial contaminants or protein aggregates)? Regarding the dialysers used for the filtration experiments, how were they cleaned after each run? Were they used only once, one module for each run (forward or backward filtration)?
The reason for selecting the specific flow rates QB and QF at the set values would also be interesting to know, as these values (and ratio between them) was different from those selected in previous published work (cited reference [25]).
The analytical methods should be better described herein or bibliographic references with full description should be indicated (spectrophotometric and chromatographic conditions; calibration methodologies).
Theoretical – Is there a reason to use the subscript “4” in the designation (and definition) of the sieving coefficient (s.c.4)? This parameter can be defined by different equations. However, since the definition is expressed in the manuscript by equation(1), the use of subscript “4” has probably a meaning. If that is the case, explain it.
Lines 116-118 – the sentence says that s.c.4 in backward filtration through asymmetric membranes should not be identical to that in forward filtration “because diameters of pores are greater outside and much smaller inside, especially those in the skin layer that is the region of 1-2 μm from the inside surface”. This causal relation does not seem right or easy to understand. The skin layer is supposed to be the selective layer and is presumed that the diameter of its pores is not changing when the flow movement through the membrane is reversed. However molecular transmission through the membrane also depends on the concentration of the solute over the selective layer, if the molecules do not stimulate pore blockage. The statement in lines 116-118 should not be in theoretical section and instead should be addressed in the discussion of results (at least mechanisms for increased s.c.4 in backward operation should be hypothesized).
Results and Discussion – Fig. 5 is already published in previous article of the authors (ref. [26]). To discuss or hypothesize the reasons for increased backfiltration s.c.4 values , images of the membranes cross-section would be relevant together with information regarding the preparation of solutions.
The inner legend of Fig. 6 (right-side graph) needs correction, as the data-points of two curves show the same open square marker. Only one curve in each graph of Fig. 6 shows standard deviations (SD). What is the reason for the lack of other SDs? If those values are so small that cannot be seen, that should be indicated in the discussion, together with indication of the order of magnitude of SD values.
The data-points in Fig. 8 (markers) should be plotted together with the curves. Are the curves perfectly aligned with the experimental values? The figure caption should also indicate that the graphs pertain to albumin transport. The same lack of data-points occurs in Figs. 8-11.
The caption in Fig. 9 should specify that the graphs pertain to the transport of dextrans.
In the discussion of Fig. 10, lines 186-189, is established a relation between the asymmetry index (IA) and the degree of membrane fouling, by comparison curves for clean and previously fouled membrane. Which are the mechanisms that cause increased asymmetry in membranes with higher fouling propensity?
Syntactic mistake – In line 179, where is written “Curves after being fouled…”, substitute by “Curves relating to fouled membranes…”
Author Response
Dear Mr. Reviewer 1;
Thank you for reviewing our manuscript entitled “Semi-quantification of the physical structure of the membrane for blood purification.” We revised our manuscript according to your comments and suggestions. Followings are answers to your questions and comments. We hope that the following answers fully dissolve your questions.
General comments:
Question#1) Why did the authors decide to change the asymmetry evaluation based on solute clearance ratio, and presently look to backward: forward filtration sieving coefficient ratio? Does the latter parameter provide better insight on asymmetric transport behaviour?
Answer#1) This membrane project originally included both dialysis experiments in which forward and backward clearances were compared and ultrafiltration experiments (this study) in which forward and backward sieving coefficients were compared. Since the former part was done earlier and may be a little easier for most clinicians to understand the concept, this part has been published before. We tried to extend our experiments in continuous basis in terms of MW rather than specific solutes of interest to draw so-called “cut-off curves” of the membrane that cannot be possible by dialysis experiments but ultrafiltration experiments with dextran with wide variety of molecular weight. This was done in this study in Figs.7 and 8, which can more directly characterize the transport characteristics and the physical structure of the membrane.
Question#2) There is a trend in the development of dual-skin hollow fibre membranes for haemodialysis (seen in, e.g., DOI: 10.1046/j.1523-1755.2000.00231.x, and DOI: 10.1016/j.aej.2022.03.043) with the purpose of gaining significant improvements in the forward transport or retention of key molecules while impairing the reverse transport of endotoxins. In light of this this trend, would it still be worth to study the membrane asymmetry parameters?
Answer#2) We know suggested articles and to the best of our knowledge, the first “dual-skin hollow fibre membrane” was developed in 1991 in Japan with polyether polymer alloy (PEPA) and it has been still used since then. We have already done most experiments on PEPA membrane in the similar manner as stated in this study and we are about to conclude that the inner skin layer had twice as high mass transfer resistances to the outer skin layer. Hopefully the manuscript about dual-skin membrane may be ready within a year or so. In conclusion, our method will work for evaluating the dual-skin membranes and is worth trying for evaluation of membranes of this kind.
Question#3) Abstract, line 23 and 30 – Specific operating conditions in the study are not what the potential readers need to know in the abstract. However, if the authors find it important, should write it in words, as the symbols used to the operating variables were not yet explained. The term “continuous” is misleading in this context and should be avoided.
Answer#3) The blood flow rate of 200 mL/min was chosen in accordance with the guideline supplied by Japanese Society for Artificial Organs (JSAO, 1982, [24]) and Japanese Society for Dialysis Therapy (JSDT, 2012, [25]) because the average blood flow rate in Japan is around 200 mL/min or even less. The same is true for the ultrafiltration rate of 10 mL/min, although this was later modified to 10 mL/min/m2-surface area in JSDT, 2012. We then added the explanation in words briefly here in “Abstract”, deleting symbols and the word “continuous.” We added more explanation in the following section, i.e., “Materials and Methods”
Revised portion related to Question#3:
L.22-L.25: Sieving coefficients (s.c.) for these solutes were measured under the test solution flow rate of 200 mL/min and the ultrafiltration rate of 10 mL/min at 310 K, according to the guideline provided by Japanese academic societies.
L.31-L.32: Above fact was verified in continuous basis by employing dextran solution before and after being fouled with albumin.
L.83-L.86: Sieving coefficients (s.c.) for those solutes were measured under the test solution flow rate QB = 200 mL/min and the ultrafiltration flow rate QF = 10 mL/min at 310 K in accordance with the guideline from Japanese Society for Artificial Organs [24] and that from Japanese Society for Dialysis Therapy [25].
Question#4) Materials and Methods – This section must be improved. Which was the quality of the water used in the preparation of solutions? Reference to the use of buffer only comes in the discussion section. Buffers were used in all or just some solutions? Which ones and for which pH values?
Answer#4) We usually do not measure the quality of water because we have previously verified that the water quality does not affect the experiments of this kind. All experiments were done with ion exchanged water that was produced from tap water with two prefilters in series. Phosphate buffer solutions were exclusively used when a protein was chosen as a test solute and all other experiments were done with ion exchanged water mentioned above. Although explanations of this question has already been shown in the bottom of Table 2, we added the explanation in the text as well.
Question#5) Were the solutions mainly binary mixtures (solute and water) or there were multicomponent mixtures of the target solutes? Which precise set of molecular dextrans were used in this work? Which was the concentration of each solute in the prepared solutions?
Answer#5) The test solution included a single component (vitamin B12, a-chymotripsin, or albumin) except for dextran that was a mixture of five chemicals with different molecular weight. None of these dextran chemicals were monodispersed and they all had wide variety of molecular weight as shown in Table 2. Initial concentrations of all the solutes were added in Table 2.
Revised portions related to Questions#4 & #5:
L104-L.111: All the transport experiments were performed aqueous in vitro with ion exchanged water produced from tap water with two prefilters in series. The test solution included a single component (vitamin B12, a-chymotrypsin, or albumin) except for dextran that was a mixture of five commercial chemicals with wide variety of molecular weight (MW), covering MW from around 1,000 to 200,000 (Table 2). A phosphate buffer solution at pH 7.4 was used as a solvent when a protein (a-chymotrypsin or albumin) was chosen as a test solute and the ion-exchanged water was used as solvent for vitamin B12 and dextran. Initial concentrations of the test solutes were also tabulated in Table 2.
Table 2 Chemicals used in the experiments |
||||
Solutes |
Molecular Weight [-] |
Produced |
Purpose |
Initial Concentration [mg/mL] |
vitamin B12 |
1,355 |
FUJIFILM Wako Pure Chemical Co., Osaka, Japan. |
Test solute |
0.025 |
a-chymotripsin1) |
25,000 |
Sigma-Aldrich, St. Louis, MO, U.S.A. |
0.305 |
|
albumin1) |
66,000 |
FUJIFILM Wako Pure Chemical Co., Osaka, Japan. |
Test solute & Foulant 2) |
24.0 |
dextran3) |
~1,500 |
Sigma-Aldrich, St. Louis, MO, U.S.A. |
Test solute |
0.504) |
~25,000 |
||||
~40,000 |
||||
~60,000 |
||||
~200,000 |
||||
1) Test solutions were prepared with phosphate buffer solution (PBS) of pH = 7.4. |
||||
2) A 2.0 % glutaraldehyde (GA) solution was used for immobilization of albumin. |
||||
3) Concentrations of dextran were measured by a gel permeation chromatography (GPC). |
||||
4) A 1.0 g of all five chemicals with different MW was dissolved in 2000 mL of the ion-exchanged water. |
Question#6) Were the test-solutions microfiltered or centrifuged at high G-force, previously to the test-runs, to eliminate possible microparticles (impurities, microbial contaminants or protein aggregates)? Regarding the dialysers used for the filtration experiments, how were they cleaned after each run? Were they used only once, one module for each run (forward or backward filtration)?
Answer#6) Test-solutions were not microfiltered or centrifuged at high G-force, previously to the test-runs because we have neither experienced aggregation of test molecules under our normal experimental conditions nor experienced to have any new peaks after finishing experiments in the spectrophotometry or in GPC. What is more, according to our calculation, possible aggregation of the test solutes would not influence significantly to the results because our interest is on the value of sieving coefficient and not on the absolute concentration. All the test dialyzers were used only once and this explanation is added in the text.
Revised portion related to Question#6:
L86-L.88: All the investigated models (dialyzers/ diafilters) were used only once and all the experiments were repeated at least three times by using separate models of the same lot.
Question#7) The reason for selecting the specific flow rates QB and QF at the set values would also be interesting to know, as these values (and ratio between them) was different from those selected in previous published work (cited reference [25]).
Answer#7) There were two reasons in the previous work for which the blood (test solution) flow rate was set less than 200 mL/min, more specifically, 100 mL/min. First of all, since the test model used in the previous work was the smallest in surface area (1.1 m2), it may be used for CRRT in which QB is usually 100 mL/min or even less. Second of all, the model included the membrane coated with vitamin E, and we had known that vitamin E may have been released from the membrane during the experiment probably due to the shear stress acting on the membrane. Since the shear stress linearly changes with the flow rate, we reduced the flow rate down to 100 mL/min to keep the intact situation of the membrane. Other than that, we make it a rule to set the blood (test solution) flow rate to be 200 mL/min. This explanation may not be suited to add into the text of this manuscript.
Revised portion related to Question#7: (none.)
Question#8) The analytical methods should be better described herein or bibliographic references with full description should be indicated (spectrophotometric and chromatographic conditions; calibration methodologies).
Answer#8) The analytical methods are added and described in detail here.
Revised portion related to Question#8:
L.114-L.126: Concentrations for vitamin B12, α-chymotrypsin, and albumin were directly measured by using an ultraviolet-visible spectrophotometer (UV-1600PC1280, Shimadzu Co., Kyoto, Japan) with the wavelength of 260, 282, and 278 nm, respectively for these solutes. and those Absorbance of the sample was measured and was converted into the concentration by using a pre-determined calibration curve, a linear correlation of the concentration as a function of the absorbance. Concentration for dextran was measured by using a gel permeation chromatography (GPC; System: 1120 Compact LC, Tosoh Co., Tokyo, Japan, Column: GF-510HQ, Showa Denko Co., Tokyo, Japan) with the flow rate of 0.5 mL/min. The mobile phase was distilled water and the volume of injected sample with no pretreatment from the automatic sampler was 30 mL. The refractive index for the differential refractometer was 64.0x10-3 RIU/FS, and the calibration curve was fit by a cubic spline.
Question#9) Theoretical – Is there a reason to use the subscript “4” in the designation (and definition) of the sieving coefficient (s.c.4)? This parameter can be defined by different equations. However, since the definition is expressed in the manuscript by equation (1), the use of subscript “4” has probably a meaning. If that is the case, explain it.
Answer#9) As far as we know, there are at least five different definitive equations of the sieving coefficient, among which s.c.3 is known to be theoretically the most accurate because this definition was derived by integrating the material balance in the infinitesimally small region in an ultrafilter, considering non-linear concentration distribution in the flow direction; however, the equation for s.c.3 is cumbersome to use and what is more it takes unstable values when it is applied to the clinical and/ or experimental data especially when the sieving coefficient is nearly equal to unity. Then we proposed a novel definition of the sieving coefficient that usually returns very close values to that of s.c.3 without showing any unstable values even under the sieving coefficient is nearly equal to unity. The new definition of ours was termed to s.c.4 by us because it is proposed after s.c.3. Explanation of this story was added in the text briefly.
Revised portion related to Question#9:
L.131-L.144: We used the sieving coefficient, s.c., as an index for permeation of the solute of interest in ultrafiltration experiments. There are, however, many definitive equations of s.c., including 1) the ratio of concentration in the filtrate to that in the blood, s.c.1 [26], 2) the ratio of concentration in the filtrate to the arithmetic mean of the concentration at the blood inlet and outlet, s.c.2 [26], and 3) the equation derived by integrating the material balance in the infinitesimally small portion of the ultrafilter, considering non-linear concentration distribution in the flow direction of the device, s.c.3 [27].  Although equations for s.c.1 and s.c.2 are simple, values from s.c.1 are always greater than those from s.c.3., while those of s.c.2 are always a little smaller than s.c.3. In addition to that, the equation for s.c.3 is a little cumbersome to use and it returns unstable values when the value is close to unity. Then we proposed a new definitive equation of the sieving coefficient in a simple way using a geometric mean of the concentration at the blood inlet and outlet in the denominator of the definition and termed it as s.c.4 [28, 29], i.e., The sieving coefficient, s.c.4, was computed by the following definition [24], i.e.,
L.148-L.151: This definition is simple, yet returns very close values to those from s.c.3 and never returns unstable values when applied to clinical and/ or experimental data even when the value is close to unity [29].
Question#10) Lines 116-118 – the sentence says that s.c.4 in backward filtration through asymmetric membranes should not be identical to that in forward filtration “because diameters of pores are greater outside and much smaller inside, especially those in the skin layer that is the region of 1-2 μm from the inside surface”. This causal relation does not seem right or easy to understand. The skin layer is supposed to be the selective layer and is presumed that the diameter of its pores is not changing when the flow movement through the membrane is reversed. However molecular transmission through the membrane also depends on the concentration of the solute over the selective layer, if the molecules do not stimulate pore blockage. The statement in lines 116-118 should not be in theoretical section and instead should be addressed in the discussion of results (at least mechanisms for increased s.c.4 in backward operation should be hypothesized).
Answer#10) This portion is just an intuitive understanding of the geometry of the wedge-like shaped pore. What we try to point out here is the difference of probability of penetration into the membrane from pores of inside surface and from those of outside surface, and do not try to argue the effect of concentration polarization formed on the membrane. Then most sentences are moved to “Results and Discussion” section with a little different expressions.
Revised portion related to Question#10:
L.152-L.157: (This paragraph is moved to L.180-L.185).
Question#11) Results and Discussion – Fig. 5 is already published in previous article of the authors (ref. [26]). To discuss or hypothesize the reasons for increased backfiltration s.c.4 values , images of the membranes cross-section would be relevant together with information regarding the preparation of solutions.
Answer#11) Fig.5 has been deleted. All the figures have been renumbered hereafter.
Revised portion related to Question#11:
L.174-L.180: Figure 5 shows the inside and outside of the hollow fiber membranes by using the FE-SEM under the magnification of 30,000. Not According to our FE-SEM observation of these membrane, not much difference between inside and outside the membrane was found in CTA (top right) that implied that it was a homogeneous membrane. We found, however, large macropores outside the ATA® (top left) and PES (bottom left) membrane with no macropores inside that proved that these membranes had asymmetric or heterogenous structures [30].
Question#12) The inner legend of Fig. 6 (right-side graph) needs correction, as the data-points of two curves show the same open square marker. Only one curve in each graph of Fig. 6 shows standard deviations (SD). What is the reason for the lack of other SDs? If those values are so small that cannot be seen, that should be indicated in the discussion, together with indication of the order of magnitude of SD values.
Answer#12) Thank you for your suggestion and we made a correction of the symbol in Fig.6 (right) – it is renumbered to Fig.5 though. We double checked the SD of these data and they are all shown in this figure, although we cannot clearly see the bars because they are too small. Along with the suggestion, we added the explanation in the text.
Revised portion related to Question#12:
L.187-L.188: All the values are represented as mean ± standard deviation (S.D.), and data are well reproduced with fairly small S.D.’s.
Question#13) The data-points in Fig. 8 (markers) should be plotted together with the curves. Are the curves perfectly aligned with the experimental values? The figure caption should also indicate that the graphs pertain to albumin transport. The same lack of data-points occurs in Figs. 8-11.
Answer#13) Data points cannot be plotted since concentrations of the dextan were measured by GPC as a continuous curve of MW, and values of s.c.4 were directly calculated from the concentrations.
Revised portion related to Question#13: (None.)
Question#14) The caption in Fig. 9 should specify that the graphs pertain to the transport of dextrans.
Answer#14) We added “using dextran solution with wide variety of MW” for the explanation of the dextran in the figure captions of Fig.8.
Revised portion related to Question#14:
Fig.8 Comparison of s.c.4 between intact and albumin immobilized membrane under forward (In-to-Out) filtration using dextran solution with wide variety of MW
Question#15) In the discussion of Fig. 10, lines 186-189, is established a relation between the asymmetry index (IA) and the degree of membrane fouling, by comparison curves for clean and previously fouled membrane. Which are the mechanisms that cause increased asymmetry in membranes with higher fouling propensity?
Answer#15) Under being fouled, the pore size of the membrane should get smaller than the original size, which decreases the forward transport of large solutes but not much the backward transport because once the molecule of interest penetrates in the pore from either inside or outside, it should go through. Therefore, the mechanism of the increased asymmetry in membranes may be caused by the reduced forward transport, leaving the backward transport nearly unchanged.
Revised portion related to Question#15:
L.239-L.244: Under being fouled, the pore size of the membrane should get smaller than the original size, which decreases the forward transport of large solutes but not the backward transport because once the molecule of interest penetrates in the pore either from inside or outside, it should go through. Therefore, the mechanism of the increased asymmetricity in membranes may be caused by the reduced forward transport, leaving the backward transport nearly unchanged.
Question#16) Syntactic mistake – In line 179, where is written “Curves after being fouled…”, substitute by “Curves relating to fouled membranes…”
Answer#16) Thank you for your comment and we modified the text according to the suggestion.
Revised portion related to Question#16:
L.225-228: Curves after being relating to fouled membranes were shifted to the left in all three membranes; moreover, much greater differences were found between intact and albumin immobilized PES (MFX) membrane than other two membranes.

Reviewer 2 Report
In this paper, the authors reported the permeation characteristics of ultrafiltration using dialysis membranes with various materials and structures. Since this paper has dealt with an interesting topic and the results have been explained by sufficient data, it is possible to publish on membranes after some modifications.
1. The notation of the reference number should be corrected. (L54) [12,13,14] -> [12-14], (L59) [17,18,19] -> [17-19], (L62) [20,21,22] -> [20 -22], (L153) [17,18,19] -> [17-19]
2. The captions of Tables and Figures should be described in more detail.
3. What is the meaning of the number 4 in the abbreviation of Sieving coefficient?
4. In Figure 6 (right), the symbols of CTA(FB) and PES(MFX) are the same, so they cannot be distinguished from each other.
Author Response
Dear Mr. Reviewer 2;
Thank you for reviewing our manuscript entitled “Semi-quantification of the physical structure of the membrane for blood purification.” We revised our manuscript according to your comments and suggestions. Followings are answers to your questions and comments. We hope that the following answers fully clear your questions.
Question#1) The notation of the reference number should be corrected. (L54) [12,13,14] -> [12-14], (L59) [17,18,19] -> [17-19], (L62) [20,21,22] -> [20 -22], (L153) [17,18,19] -> [17-19]
Answer#1) Thank you for your suggestion. We changed the format of the reference number according to the suggestion.
Question#2) The captions of Tables and Figures should be described in more detail.
Answer#2) We reconsidered and changed almost all the captions of Tables and Figures more informative.
Question#3) What is the meaning of the number 4 in the abbreviation of Sieving coefficient?
Answer#3) As far as we know, there are at least five different definitive equations of the sieving coefficient, among which s.c.3 is known to be theoretically the most accurate because this definition was derived by integrating the material balance in the infinitesimally small region in an ultrafilter, considering non-linear concentration distribution in the flow direction; however, the equation for s.c.3 is cumbersome to use and what is more it takes unstable values when it is applied to the clinical and/ or experimental data especially when the sieving coefficient is nearly equal to unity. Then we proposed a novel definition of the sieving coefficient that usually returns very close values to that of s.c.3 without showing any unstable values even under the sieving coefficient is nearly equal to unity. The new definition of ours was termed to s.c.4 by us because it is proposed after s.c.3. Explanation of this story was added in the text.
Revised portion related to Question#3:
L.132-L.145: We used the sieving coefficient, s.c., as an index for permeation of the solute of interest in ultrafiltration experiments. There are, however, many definitive equations of s.c., including 1) the ratio of concentration in the filtrate to that in the blood, s.c.1 [26], 2) the ratio of concentration in the filtrate to the arithmetic mean of the concentration at the blood inlet and outlet, s.c.2 [26], and 3) the equation derived by integrating the material balance in the infinitesimally small portion of the ultrafilter, considering non-linear concentration distribution in the flow direction, s.c.3 [27].  Although equations for s.c.1 and s.c.2 are simple, values from s.c.1 are always greater than those from s.c.3., while those of s.c.2 are always a little smaller than s.c.3. In addition to that, the equation for s.c.3 is a little cumbersome to use and it returns unstable values when the value is close to unity. Then we proposed a new definitive equation of the sieving coefficient in a simple way using a geometric mean of the concentration at the blood inlet and outlet in the denominator of the definition and termed it as s.c.4 [28, 29], i.e., The sieving coefficient, s.c.4, was computed by the following definition [24], i.e.,
L.149-L.152: This definition is simple yet returns very close values to those from s.c.3 and never returns unstable values when applied to clinical and/ or experimental data even when the value is close to unity.
Question#4) In Figure 6 (right), the symbols of CTA(FB) and PES(MFX) are the same, so they cannot be distinguished from each other.
Answer#4) Thank you for your comment. We corrected the error according to your suggestion.

Round 2
Reviewer 1 Report
Dear authors,
Below, please find a few comments/suggestions regarding the revised version of your manuscript.
Line 180 – The start of the sentence “As shown in Fig.4...” should perhaps be altered, because Fig.4 does not show s.c.4 values; Fig.5 does.
New Fig.5 – The revised figure (former Fig.6) is now correct but became a little fuzzy.
Figs.5 to 10 – Could their definition still be improved? These figures were resized which is convenient for manuscript compactness but, like in new Fig.5, the resolution of the images has been reduced and that is not so good.